# Chemical signal activation of an organocatalyst enables control over soft material formation

Fanny Trausel[1], Chandan Maity[1], Jos M. Poolman[1], D.S.J. Kouwenberg[1], Frank Versluis[1], Jan H. van Esch [1] & Rienk Eelkema [1]

Cells can react to their environment by changing the activity of enzymes in response to specific chemical signals. Artificial catalysts capable of being activated by chemical signals are rare, but of interest for creating autonomously responsive materials. We present an organocatalyst that is activated by a chemical signal, enabling temporal control over reaction rates and the formation of materials. Using self-immolative chemistry, we design a deactivated aniline organocatalyst that is activated by the chemical signal hydrogen peroxide and catalyses hydrazone formation. Upon activation of the catalyst, the rate of hydrazone formation increases 10-fold almost instantly. The responsive organocatalyst enables temporal control over the formation of gels featuring hydrazone bonds. The generic design should enable the use of a large range of triggers and organocatalysts, and appears a promising method for the introduction of signal response in materials, constituting a first step towards achieving communication between artificial chemical systems.

---

[1] Department of Chemical Engineering, Delft University of Technology, van der Maasweg 9, Delft 2629 HZ, The Netherlands. Correspondence and requests for materials should be addressed to R.E. (email: r.eelkema@tudelft.nl)

C ontrol over enzymatic activity is at the basis of cellular communication and the regulation of a wide range of biological processes. Enzymes are often activated in a covalent manner by (de-)phosphorylation, for example in cell signalling. Allosteric enzymes, such as haemoglobin, are hindered or stimulated by non-covalent interactions with a regulatory molecule[1, 2]. Such rudimentary forms of communication and regulation are almost entirely absent in artificial materials, but could lead to the development of soft materials capable of autonomously responding to changes in their environment. Although enzymes are typically very efficient catalysts, the type of reactions they catalyse and their operating conditions are limited. An artificial catalyst that is activated by a chemical signal opens possibilities for autonomous spatial and temporal control over systems at a molecular and supramolecular level. Only a few examples of synthetic catalysts with addressable activity exist, predominantly controlled using light as a signal[3–7]. Catalysts whose activity is controlled using a chemical signal are nearly absent from the literature[8–11] and often suffer from complex design and synthesis as well as a very specific operating mechanism.

Here, we present proof of principle of an organocatalyst that is activated by a chemical signal. This concept provides a generic design to enable autonomous response to biologically and mechanically relevant signals from the environment. The signal responsive catalyst is designed as a self-immolative molecule[12–15], which fragments upon reaction with a specific chemical signal to release an active catalyst. We synthesise a protected aniline (pro-aniline 1) that liberates the organocatalyst aniline 2 upon reaction with the chemical signal $H_2O_2$. Aniline is a nucleophilic catalyst for hydrazone formation and exchange[16–18], a reaction frequently used in soft[19] and dynamic covalent materials[20]. $H_2O_2$ is released

by the enzymatic oxidation of many different disease related biomarkers such as glucose, lactose, sarcosine, uric acid, choline and acetylcholine, making it a highly relevant biological signal[21–23]. Furthermore, strained polymers have been shown to generate $H_2O_2$ in the presence of water, making $H_2O_2$ an interesting mechanically generated chemical signal[24]. Activating pro-aniline 1 with $H_2O_2$ as a chemical signal leads to an almost instant 10-fold increase in rate of hydrazone formation. In addition, using the pro-catalyst in gels featuring hydrazone bonds enables control over material formation, creating materials that can respond to a specific chemical signal.

## Results

**Activation of the catalyst.** The pro-catalyst pro-aniline 1 is activated by the chemical signal $H_2O_2$ and catalyses hydrazone formation (Fig. 1a, b). Pro-aniline 1 was synthesised in only two steps in good yields (Fig. 1c). We confirmed the release of aniline 2 from pro-aniline 1 (72 mM) upon reaction with $H_2O_2$ (18 equivalents) using GC/MS (gas chromatography/mass spectrometry), showing that 1 fragments completely to release 2 (Supplementary Fig. 1). We analysed the kinetics of aniline 2 release using UV/vis spectroscopy (Supplementary Fig. 2a, b). The timescale for the liberation of 2 from 1 (0.5 mM) depends on the amount of $H_2O_2$ added, taking 5 min using 50 equivalents of $H_2O_2$, and 15 min using 10 equivalents. In the absence of the $H_2O_2$ signal, pro-aniline 1 is stable in solution for over 15 h (Supplementary Fig. 2c, d).

We investigated the catalytic activity of pro-aniline 1 upon activation, by comparing the rates of hydrazone formation in a model reaction (Fig. 2a)[25, 26]. Aldehyde 4 (0.5 mM) reacts with hydrazide 3 (0.1 mM) to form hydrazone 5 in a buffered medium

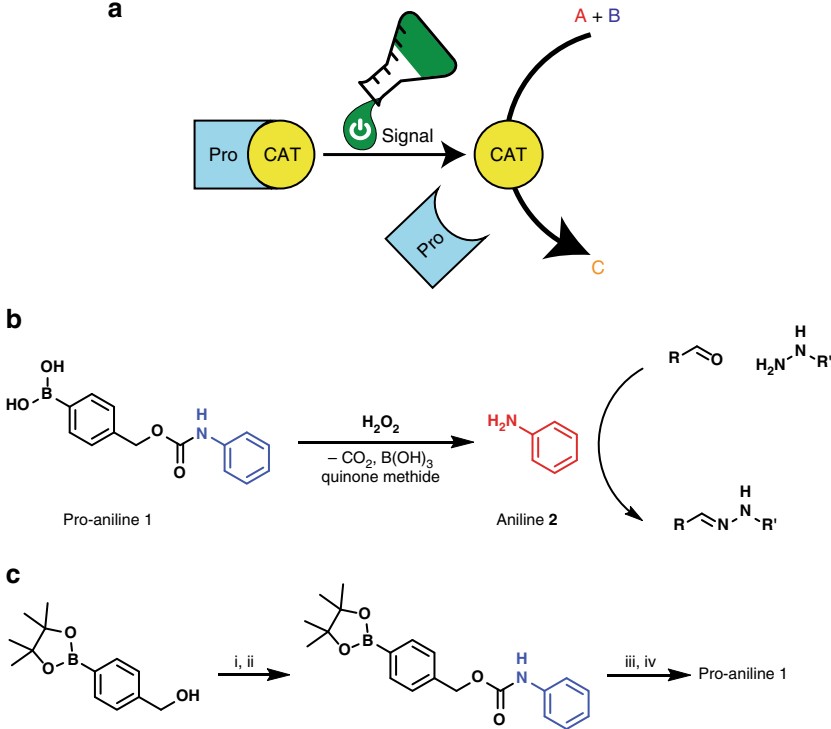

**Fig. 1** Design and synthesis of the protected catalyst. **a** Schematic representation of the activation of a protected catalyst and subsequent catalysis of a chemical reaction. **b** The pro-catalyst pro-aniline 1 and the chemical signal $H_2O_2$ react to release the organocatalyst aniline 2, which catalyses hydrazone formation between an aldehyde and a hydrazide. **c** Synthetic route for the synthesis of pro-aniline 1, (i) $K_2CO_3$, triphosgene, 0 °C – room temperature, (ii) aniline 2, $NaHCO_3$, tetrahydrofuran, 0 °C – room temperature, (iii) $NaIO_4$, ammonium acetate, room temperature, (iv) 1 M aqueous HCl. Total yield over two steps is 64%

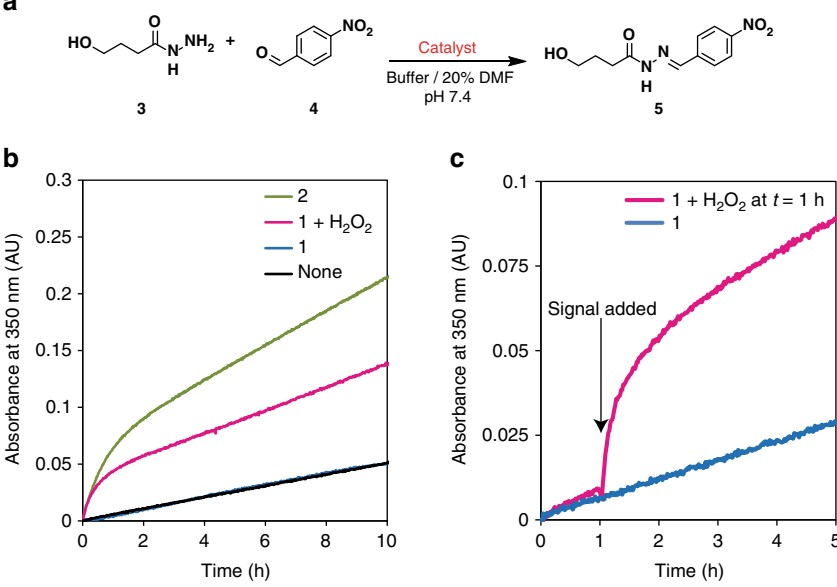

**Fig. 2** Control over the rate of hydrazone formation by activation of the catalyst. **a** Model hydrazone formation reaction, yielding the UV-active probe **5**. Reaction conditions: 0.1 mM hydrazide **3**, 0.5 mM aldehyde **4**, 0.5 mM pro-aniline **1**, 0.5 mM aniline **2**, 2.5 mM (5 equivalents) $H_2O_2$ in 20% dimethylformamide (DMF) in phosphate buffer (100 mM, pH 7.4). All experiments were carried out at 25 °C. The 20% DMF was used to ensure that **1** was completely dissolved. The rate of formation of **5** is 10 times higher when using $H_2O_2$-activated pro-aniline **1** than for unactivated pro-aniline **1**. **b** Formation of **5** over time, without catalyst (black line), in the presence of aniline **2** (green line), in the presence of unactivated pro-aniline **1** (blue line) and in the presence of activated pro-aniline **1** with 5 equivalents of $H_2O_2$ (magenta line). **c** The rate of formation of **5** can be controlled during the process by adding a chemical signal (5 equivalents of $H_2O_2$ here after 1 h), liberating the catalyst. Reaction with pro-aniline **1** and subsequent addition of $H_2O_2$ (magenta line), reaction with unactivated pro-aniline **1** (blue dashed line). After addition of the chemical signal an immediate response was observed: the reaction rate increased nine-fold

### Table 1 Activation of the catalyst determines the initial rate of hydrazone formation

| Catalyst system | $k_1$ ($10^{-6}$ s$^{-1}$) | $k_{rel}$ |
|---|---|---|
| none | 6.1 ± 0.1 | 1.0 |
| $H_2O_2$ | 6.0 ± 0.3 | 1.0 |
| **1** | 5.6 ± 0.7 | 0.9 |
| **1** + $H_2O_2$ | 60 ± 8 | 10 |
| **2** | 113 ± 5 | 19 |

Pseudo-first-order reaction rate constants ($k_1$) for hydrazone formation were determined by following the absorbance of **5** in UV/vis spectroscopy. The errors are the standard error of mean (the standard deviation divided by the square root of the number of measurements)

(100 mM phosphate buffer pH 7.4) with 20% dimethylformamide (DMF) (Fig. 2b). This reaction is catalysed by aniline **2** (0.5 mM), giving a 19-fold increase in reaction rate with respect to the uncatalyzed reaction. Unactivated pro-aniline **1** (0.5 mM) does not influence the reaction rate of hydrazone formation. Addition of $H_2O_2$ (2.5 mM) to pro-aniline **1** (0.5 mM) gives a relative reaction rate of 10, indicating efficient activation of the organocatalyst (Fig. 2b and Table 1). $H_2O_2$ alone (2.5 mM) does not increase the reaction rate of hydrazone formation (Supplementary Fig. 3a, b). Furthermore, the pH was monitored during the reaction with pro-aniline **1** and $H_2O_2$: the pH remained stable at a value of 8. As our solvent system alone (100 mM phosphate buffer with 20% DMF) gives a pH of 8, we conclude that the reactions were sufficiently buffered. After activation of pro-aniline **1**, the reaction rate is lower than when using native aniline **2**. In an attempt to explain this apparent loss of catalytic activity, we investigated the influence of $H_2O_2$ and of boric acid on the activity of aniline **2**, but did not observe a lower rate (Supplementary Fig. 3c, d). Although we confirmed complete conversion of pro-aniline **1** after addition of more than 1

equivalent of $H_2O_2$ and we were able to detect aniline **2** after an overnight hydrazone reaction in the presence of a substoichiometric amount of pro-aniline **1** + $H_2O_2$ (Supplementary Fig. 4), it might be the case that a small amount of aniline **2** is degraded over time. Importantly, using our signal responsive catalyst, we should be able to elicit a change in reaction rate at any given time during the process. To show this, pro-aniline **1** (0.5 mM) and reactants **3** (0.1 mM) and **4** (0.5 mM) were dissolved in buffer and mixed. Upon addition of the signal $H_2O_2$ after 1 h, we observed an immediate 9-fold increase in the reaction rate (Fig. 2c). The rapid and significant increase of reaction rate when exposed to a chemical signal shows that the activation of a pro-catalyst enables an instant and autonomous response to a chemical change in the environment.

**Control over gel formation by activation of the catalyst**. As a first application of the signal responsive catalyst, we chose to couple the formation of a hydrazone polymer gel material[27] to a chemical signal. We synthesised an alternating polyethylene glycol/benzaldehyde copolymer using mesylated polyethylene glycol (molecular weight 5.4–6.6 kg mol$^{-1}$) and 3,4-dihydroxybenzaldehyde. A polydisperse polymer **6** ($M_w$ ~ 1×10$^5$ g mol$^{-1}$) featuring benzaldehyde groups was obtained (Fig. 3a). Hydrazone formation between polymer aldehyde **6** and trishydrazide **7** is catalysed by aniline **2** and leads, under the right conditions, to the formation of transparent polymer gels (Fig. 3a, c). Using the signal induced activation of pro-aniline **1** we are able to control the rate of gel formation, the moment of gel formation (temporal control), and the mechanical properties of the obtained gels. An inverted tube test was used to investigate the influence of catalyst activation on gel formation. The gel tests presented here were performed using 136 mg ml$^{-1}$ polymer aldehyde **6**, 10 mM hydrazide **7** and 10 mM catalyst in 20% DMF in aqueous buffer

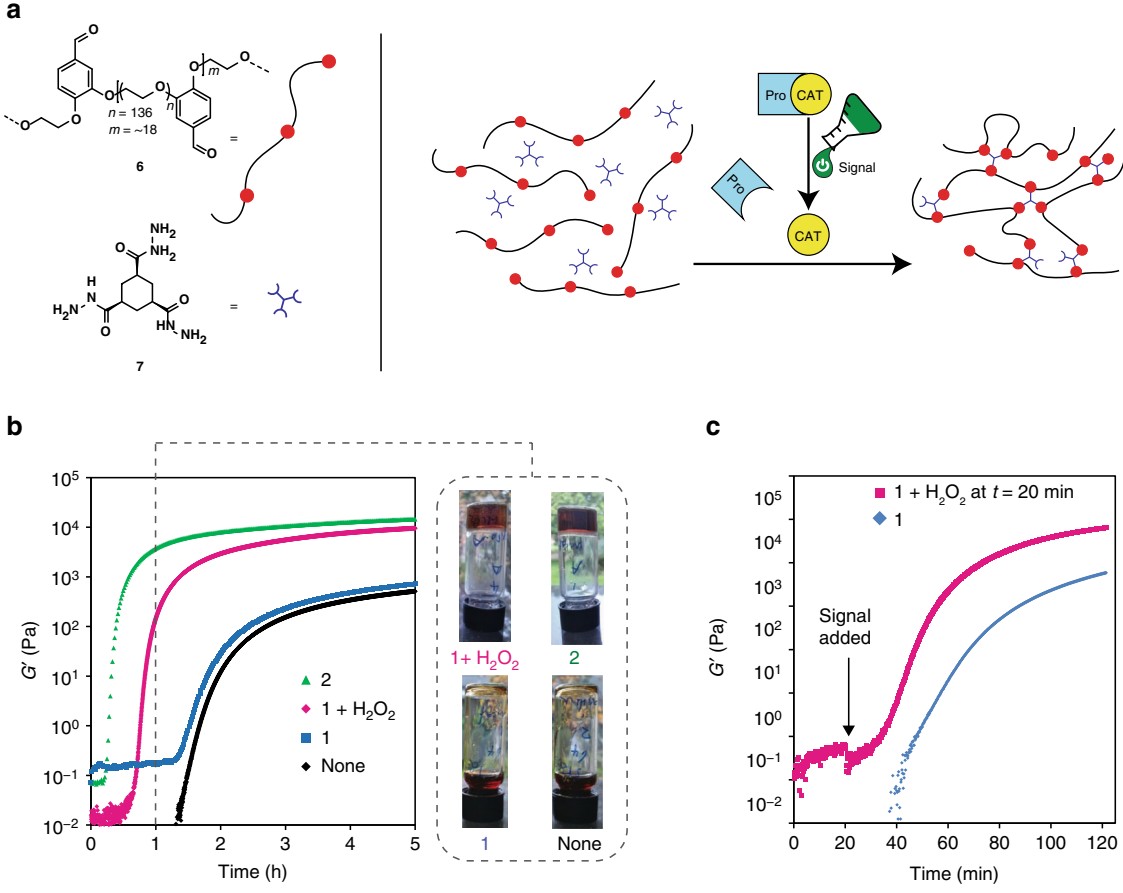

**Fig. 3** Polymer gel formation controlled by the activation of pro-aniline **1**. **a** The formation of a hydrazone polymer gel: aldehyde **6** and hydrazide **7** react to form a crosslinked hydrazone polymer gel. **b** The storage modulus $G'$ measured with oscillatory rheology during gel formation at 25 °C (10 mM hydrazide **7**, 136 mg ml$^{-1}$ aldehyde **6** in 20% dimethylformamide in 100 mM phosphate buffer, pH 6.0) for the uncatalyzed gel (black line), the gel with 10 mM pro-aniline **1** (blue line), the gel with 10 mM pro-aniline **1** and 10 equivalents $H_2O_2$ (magenta line) and the gel with 10 mM aniline **2** (green line). The gelation rate with activated pro-aniline **1** is comparable to the gelation rate with aniline **2**. Without activation of the pro-aniline **1**, the gelation rate is comparable to the gelation rate of the uncatalyzed reaction. After 1 h of gelation time, the mixtures with aniline **2** or activated pro-aniline **1** have gelled, whereas the mixtures without catalyst or with unactivated pro-aniline **1** have not gelled yet. We performed the gelation experiment in vials using the same conditions as we used for rheology and took a photograph after 1 h of reaction time. Top left: pro-aniline **1** (10 mM) and $H_2O_2$ (100 mM), top right: aniline **2** (10 mM), bottom left: pro-aniline **1** (10 mM) and bottom right: without catalyst. **c** The gelation rate can be controlled during the process by adding a chemical signal (here after 20 min), liberating the catalyst. The storage modulus $G'$ measured with oscillatory rheology for the aldehyde **6**/hydrazide **7** (10 mM) mixture with 10 mM pro-aniline **1** (blue line) and the mixture with pro-aniline **1** when 10 equivalents of $H_2O_2$ was added after 20 min (magenta line). Addition of the chemical signal induces a significant increase in gelation rate

(100 mM phosphate buffer pH 7.4) in 4 ml vials. The hydrazide **7** and aldehyde **6** mixtures with either aniline **2** or pro-aniline **1** activated with $H_2O_2$ (10 equivalents) form gels within 1 h (Fig. 3b). In contrast, for the mixtures with unactivated pro-aniline **1** or without catalyst, gelation takes 2 h.

We performed oscillatory rheology to quantify the rate of gel formation under influence of the catalyst and to investigate the mechanical properties of the formed materials (Fig. 3b). The gel prepared with unactivated pro-aniline **1** is comparable in stiffness and formation time to the uncatalyzed gel, whereas the gel prepared with pro-aniline **1** and $H_2O_2$ (10 equivalents) is comparable to the gel formed using aniline **2** as a catalyst. After 6 h of gelation time the catalysed gels show elastic moduli ($G'$) that are 1.5 times higher than the $G'$ values we measured for the uncatalyzed gels, which indicates that the gel stiffness is controlled by catalysis[19, 28]. The difference in timescale of gel formation is especially apparent in rheology, as the cross-over for $G'$ and $G''$ (the gel point) for the catalysed gels is observed after 30 min, whereas this cross-over takes place after almost 2 h for

uncatalyzed gels (Supplementary Fig. 5a). Thus, activation of the catalyst by the chemical signal influences the gel formation rate as well as the gel stiffness.

With the signal responsive catalyst, we can now attempt to control the moment of material formation using a chemical signal. In the rheometer, we added a chemical signal to a solution of **6** and **7** containing pro-aniline **1**, 20 min after mixing all components (Fig. 3c). We observed a significant increase in the rate of gel formation shortly after addition of the chemical signal. A control experiment lacking the added signal showed a delayed and smaller increase of the elastic modulus. Importantly, these observations show that our system allows for temporal control over reaction rates and material formation.

**Control over supramolecular gel formation by activation of the catalyst.** To investigate the scope of our signal responsive catalyst, we also used pro-aniline **1** to control the formation of a supra-molecular trishydrazone hydrogel described previously by our

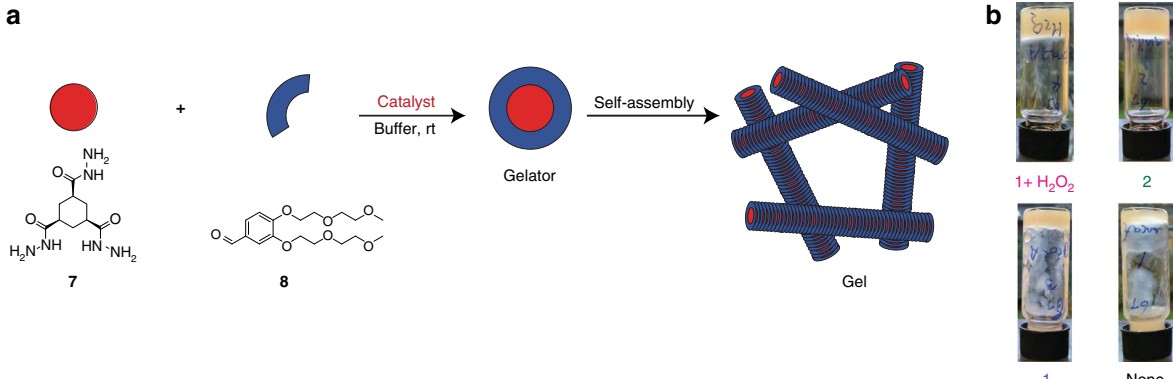

**Fig. 4** Supramolecular gel formation controlled by the activation of pro-aniline **1**. To demonstrate the versatility of the pro-aniline system, we used the pro-aniline **1** to control the formation of a supramolecular gel[18]. **a** Schematic representation of the two water-soluble building blocks aldehyde **8** and hydrazide **7** that react by hydrazone formation in the presence of a catalyst to form a gelator, which stacks into fibres and forms a gel. **b** We performed a gelation test for the supramolecular gel in vials. Photographs of gelation samples taken after 15 h of reaction time for the mixture (16 mM hydrazide **7** and 96 mM aldehyde **8** in 20% methanol in phosphate buffer, 100 mM, pH 6.0) with, top left: pro-aniline **1** (10 mM) and $H_2O_2$ (100 mM), top right: aniline **2** (10 mM), bottom left: unactivated pro-aniline **1** (10 mM), bottom right: without catalyst. The mixtures with aniline **2** or pro-aniline **1** with $H_2O_2$ formed gels after 15 h, whereas the mixture without catalyst or with unactivated pro-aniline **1** did not form gels, although the viscosity of the mixtures was increased after 15 h

group[19, 29], featuring trishydrazide **7** (16 mM) and aldehyde **8** (96 mM) (Fig. 4a). Using either aniline **2** (10 mM), or pro-aniline **1** (10 mM) with $H_2O_2$ (10 equivalents), the supramolecular gels forms overnight. In contrast, without catalyst or with unactivated pro-aniline **1**, no gels are obtained (Fig. 4b). The activated pro-aniline can thus be used to control the formation rate and properties of polymer gels as well as supramolecular gels featuring hydrazone bonds.

## Discussion

In summary, we report a deactivated organocatalyst that is activated by a chemical signal. The pro-aniline **1** was synthesised in only two steps in good yields. Activating the pro-aniline **1** organocatalyst with $H_2O_2$ as a chemical signal increases the rate of hydrazone formation 10-fold almost instantly. We obtained control over gel formation for a polymer gel and for a supramolecular gel, both featuring hydrazone bonds. For the polymer gel, we controlled the moment of gelation: addition of the chemical signal to the unactivated pro-aniline **1** catalyst at an arbitrary moment after mixing all gelation components leads to a significant increase in the rate of gel formation. This shows that it is possible to control the moment of material formation in response to a chemical signal. As self-immolative trigger groups allow response to a wide range of signals[12], we have aimed to develop a generic method for the design of activatable catalysts. Furthermore, since self-immolative molecules can be used to incorporate more than one molecule of interest, the design can be used for signal amplification[12]. We are currently working on several organocatalysts that are responsive to a range of specific chemical signals. With the current results, we have developed a promising method for the introduction of signal response in molecular materials, constituting a first step towards achieving communication between artificial chemical systems.

## Methods

**Instrumentation and characterisation.** Nuclear magnetic resonance (NMR) spectra were recorded on an Agilent-400 MR DD2 (400 MHz for $^1$H and 100.5 MHz for $^{13}$C) at 298 K using residual protonated solvent signals as internal standard ($^1$H: δ(CHCl$_3$) = 7.26 p.p.m., δ(CH$_3$OH) = 3.31 p.p.m. and $^{13}$C: δ(CHCl$_3$) = 77.16 p.p.m., δ(CH$_3$OH) = 49.00 p.p.m.). Thin layer chromatography (TLC) was performed on Merck Silica Gel 60 F254 TLC plates with a fluorescent indicator with a 254 nm excitation wavelength and compounds were visualised under UV light of 254 nm wavelength. GC/MS was performed with a Shimadzu QP-2010S

GC/MS. Liquid chromatography–mass spectrometry (LC/MS) was performed on a Shimadzu Liquid Chromatograph Mass Spectrometer 2010, LC-8A pump with a diode array detector SPD-M20. The column used was the Xbridge Shield RP 18.5 μm (4.6 × 150 mm). UV/Vis spectroscopic measurements were performed on an Analytik Jena Specord 250 spectrophotometer; quartz cuvettes with a path length of 1 cm were used. Oscillatory experiments were performed using an AR G2 rheometer from TA Instruments in a strain-controlled mode; the rheometer was equipped with a steel plate-and-plate geometry of diameter 40 mm and a water trap. The temperature of the plates was controlled at 25 ± 0.2 °C. Measurements were performed at a frequency of 1 Hz while applying 1% strain. Gel permeation chromatography (GPC) was performed using a Shimadzu Prominence GPC system equipped with 2× PL aquagel-OH MIXED H columns (Agilent, 8 μm, 300 × 7.5 mm) and refractive index detector.

**Synthetic procedures.** For the synthesis of *cis,cis*-cyclohexane-1,3,5-tricarbohydrazide **7** and 3,4-bis(2-(2-methoxyethoxy)ethoxy)benzaldehyde **8** we refer to procedures described in the literature[30]. Detailed procedures for the synthesis of pro-aniline **1**, hydrazone **5** and PEG-aldehyde copolymer **6** are described in the Supplementary Methods, NMR and GPC spectra are shown in Supplementary Figs 9–17.

**Investigation of the release of aniline 2 from pro-aniline 1.** The self-immolative reaction of pro-aniline **1** with $H_2O_2$ was investigated with TLC, GC/MS and UV/vis spectroscopy. TLC: pro-aniline **1** (4.8 mg, 13.6 μmol) was dissolved in acetonitrile (0.5 ml) and a solution of $H_2O_2$ in deionized water (18 equivalents, 0.245 mM, 0.5 ml) was added. TLCs (eluent = 1:1 petroleum ether: ethyl acetate) were taken at $t$ = 0, 1, 5, 10, 15 and 20 min. The reaction mixture and aniline **2** were run on the TLC plate for comparison. After 10 min all starting compound had disappeared, replaced by aniline **2** and side products. GC/MS: pro-aniline **1** (5 mg, 18 μmol) was dissolved in 0.25 ml ethyl acetate. $H_2O_2$ (18 equivalents, 25.4 μl in 0.25 ml deionized water) was added and after the mixture was allowed to stir for 1 h at room temperature, the reaction was quenched with a saturated sodium thiosulfate solution. The organic layer of the reaction mixture was analysed by GC/MS: after the reaction the product peak had disappeared and aniline **2** ($m/z$ 93) was detected in the reaction mixture (Supplementary Fig. 1). UV/vis: the self-immolative reaction of pro-aniline **1** followed in UV/vis was performed in a 20% DMF in phosphate buffer (100 mM, pH 7.4) solution, using 0.5 mM concentration of pro-aniline **1** and 18 equivalents of $H_2O_2$ (Supplementary Fig. 2a, b)). The self-immolative reaction is complete when the increase of absorption reaches a plateau. The stability measurement of pro-aniline **1** overnight was performed using the same conditions of pro-aniline **1** (0.5 mM) in a mixture of 20% DMF in buffer. Without $H_2O_2$ pro-aniline **1** is stable for over 15 h (that is, does not release aniline **2**, Supplementary Fig. 2c, d).

**Investigation of the catalytic activity of pro-aniline 1 after activation.** A calibration line was measured for hydrazone product **5** in 20% DMF in phosphate buffer (100 mM, pH 7.4). The extinction coefficient of hydrazone **5** at 350 nm under these conditions is $1.3 \pm 0.087 \times 10^4$ $M^{-1}$ $cm^{-1}$ (Supplementary Fig. 6). The hydrazone reaction (Fig. 2a) was performed in 20% DMF in a 100 mM phosphate buffer pH 7.4, containing 0.5 mM of 4-nitrobenzaldehyde **4**, 0.5 mM of catalyst, 0.1

mM of 4-hydroxybutyric acid hydrazide **3** and 2.5 mM of $H_2O_2$. The quartz cuvettes contained a total reaction volume of 2 ml. The stock solutions of the reagents were added as follows: aldehyde **4** solution (10 mM in DMF), phosphate buffer, DMF, catalyst solution (10 mM in DMF), the hydrazide **3** solution (2 mM in buffer). The cuvettes were closed using Teflon caps and thoroughly mixed by converting the cuvette upside down four times. The spectra of the starting reaction mixtures were measured (reference measurement, 250–450 nm, 10 nm s$^{-1}$). The product peak at 350 nm was followed using a 6-sample holder (standard absorption measurement, slow time scan, measuring wavelength 350 nm, scan every 60 s, Fig. 2b, c). At the end of the measurement, single scans were measured again using the same settings as for the starting reaction mixtures (Supplementary Fig. 7). The pH was monitored for the reaction with pro-aniline **1** (0.5 mM) and $H_2O_2$ (2.5 mM), every 10 min for the first 2 h and once after 18 h. At all times a pH of 8.0 was measured. As the phosphate buffer (100 mM, pH 7.4) with 20% DMF alone also gives a pH of 8.0, this indicates that the pH did not change during the reaction. We ensured that no side reactions occurred (Supplementary Table 1). Addition of aniline **2** to the uncatalyzed reaction after 1 h of reaction time gave a similar response in reaction rate as addition of $H_2O_2$ to the reaction with pro-aniline **1** (Supplementary Fig. 3e). The pseudo-first-order rate constants were determined by converting the absorbance measured during the reactions to concentration using the extinction coefficient of the product and by fitting the natural logarithm of the concentration (M) over time (s). The graph was fitted using linear regression in Origin Pro 2015 to yield the pseudo-first-order reaction rate constant. For the uncatalyzed reaction and the reaction with pro-aniline **1**, the first 10 h of the reaction were taken to determine the rates. For the reaction in the presence of activated pro-aniline **1** and for the reaction catalysed by aniline **2**, the first 15 min were used to determine the rate (Supplementary Fig. 8 and Supplementary Table 2). To ensure that aniline **2** was not used up during the reaction, we performed a 65 h reaction of hydrazide **3** with aldehyde **4** in the presence of 0.5 equivalents of pro-aniline **1** with $H_2O_2$ and analysed the reaction mixture afterwards with GC/MS. We were able to detect aniline **2** (*m/z* 93) in the reaction mixture. Conditions: hydrazide **3** (20 mM), aldehyde **4** (20 mM), pro-aniline **1** (10 mM), $H_2O_2$ (11 mM) in 20% DMF in 100 mM phosphate buffer pH 7.4. The reaction mixture was extracted after 65 h of reaction time with dichloromethane. The organic layer was evaporated and re-dissolved in ethyl acetate for GC/MS analysis. Aniline **2** was detected in the reaction mixture, MS (GC/MS) *m/z*: 93 [M], 66 $[C_5H_6]^{+\bullet}$ (expected *m/z* = 93.06), retention time: 11 min.

**Rheology of the polymer gel**. The storage and loss moduli $G'$ and $G''$ were followed in time during the formation of the gel, using a rheometer (Supplementary Fig. 5a). Total volume of the gels is 0.6 ml. Composition of the gels: 140 mg ml$^{-1}$ polymer aldehyde **6**, 10 mM aniline **2** or 10 mM pro-aniline **1**, 10 mM hydrazide **7**, 20% DMF in 100 mM phosphate buffer pH 6.0. First, the polymer aldehyde **6** (84 mg) was weighed out in the shell vial, then we added buffer, the catalyst solution (120 μl of 50 mM stock solution in DMF) or 120 μl DMF, the $H_2O_2$ solution (120 μl of 500 mM stock solution in buffer, 10 equivalents) and last, the hydrazide **7** solution (150 μl of 40 mM stock solution in buffer). The mixture was vortexed and poured directly on the rheometer plate. The rheometer plate was rotated slowly when it was lowered to ensure equal spreading of the sample. The $G'$ and $G''$ were measured while the gel formed on the plate. When no significant change in the moduli was observed anymore, the measurement was stopped. The frequency dependence of the gels was measured (Supplementary Fig. 5b): none of the gels showed frequency dependency in the frequency range that was used (0.01–100 Hz). A strain sweep was measured additionally; a strain higher than 100% could usually not be applied, as the gel gave too much resistance to the rheometer (Supplementary Fig. 5c). At increasing strain, the oscillatory stress increases linearly (Supplementary Fig. 5d). The rheology experiments where the chemical signal was added during the measurement, a gel volume of 1 ml was used. After mixing the stock solutions of polymer aldehyde **6**, hydrazide **7** and pro-aniline **1**, we poured the mixture on the rheometer plate and allowed gelation while measuring rheology. After 20 min the measurement was stopped, the top rheometer plate was raised and $H_2O_2$ (50 μl, 2 M, 10 equivalents) was added (Fig. 3c). We also performed an experiment where we let the uncatalyzed gel form and added aniline **2** (50 μl, 200 mM) after 60 min (Supplementary Fig. 5e).

**Inverted tube test supramolecular gel**. Aldehyde **8**, hydrazide **7** and catalyst solutions were mixed in 4 ml vials and left overnight. The next day, the vials were turned upside down. When the content of the vial would stay at the bottom of the vial for at least 5 min and was able to sustain its own weight, we assumed a gel had formed. Gelation conditions: 16 mM *cis,cis*-cyclohexane-1,3,5-tricarbohydrazide **7**, 96 mM 3,4-bis(2-(2-methoxyethoxy)ethoxy)benzaldehyde **8**, 10 mM catalyst, 100 mM $H_2O_2$, solvent 20% methanol in phosphate buffer (100 mM, pH 6.0). Sample size = 1 ml. The gelation took place overnight at room temperature (Fig. 4a, b).

**Data availability**. Data relevant to the findings of this study are available from the corresponding author on request.

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

## Acknowledgements

This work was supported by the Netherlands Organisation for Scientific Research (NWO VIDI grant to R.E.). We thank Maarten C.J.K. Gorseling for GC/MS measurements.

## Author contributions

F.T. and R.E. conceived the research. R.E. directed the research. F.T. carried out the experiments. C.M. synthesised pro-aniline **1**, J.M.P. designed the polymer aldehyde **6**, D.S.J.K. synthesised polymer aldehyde **6**, F.V. and R.E. revised the manuscript, J.H.v.E. provided suggestions on experiments and improvements. F.T. wrote the manuscript, all authors commented on the work and the manuscript.

## Additional information

**Competing interests:** The authors declare no competing financial interests.

