## [Peer Review File · Nature Communications]

Reviewers' comments:

Reviewer #1 (Remarks to the Author):

This is an interesting document which reports the use of a chemically activated catalyst to mediate a reaction of interest between aldehyde and hydrazine. The authors propose that this then constitutes a method by which temporal control of materials performance can be achieved using a chemical trigger. The chemical trigger chosen in this case is hydrogen peroxide which is of biological relevance – the authors demonstrate that polymer and supramolecular gel-type materials can be formed in response to this trigger.

The biggest issue with the manuscript is that, taken on its own the product of the H₂O₂-mediated reaction (compound 2) appears to be far less active as a catalyst (10 times) than the system formed by mixing its precursor (1) with H₂O₂. As such, it cannot really be concluded that compound 2 alone is responsible for the increase in reaction efficiency. The authors are up-front about this and state they cannot think of reasons why this may be the case.

I would like to see the authors outline more clearly why the aniline can act as a catalyst in the first instance – this is never clearly stated in the paper. This would help further considerations.

The authors should then think carefully about the oxidising potential of hydrogen peroxide – does it oxidise the amine 2, or form a complex with it – changing the nature of the catalyst in some way? Or does H₂O₂ play some other role in mediating the reaction between aldehyde and hydrazine (after all it does appear, on its own, to have a catalytic effect on the reaction).

Alternatively, does hydrogen peroxide change the pH of the medium? Were the reactions buffered, or pH controlled in some other way. All of these factors need considering before the authors could publish. I accept the problem may not be solvable, but it is important that the authors rule out as many possibilities as they can, and have some sort of proposal as to why they are observing this strange effect.

The authors refer consistently to compound 2 as a 'catalyst'. Yet it is used in almost all examples at stoichiometric (or super-stoichiometric) levels. Often one of the other reagents is the limiting factor. Is it correct to strictly refer to it as a catalyst?? A catalyst must turn over multiple reactions, and I am not sure I see evidence that it does. Again, if the authors were clear about the mechanism of aniline-mediated catalysis of these reactions, this may become clearer.

The authors apply the system in materials synthesis. I note that the difference made by the chemical trigger in the formation of the polymer gel is relatively small – after all, in the absence of hydrogen peroxide the gel still forms. As such, this system can only be seen as a first step towards what the authors are aiming for. I think they should perhaps describe what they observe as 'partial temporal control' – it is clear that the gelation is potentiated by the addition of the signal, but it would have happened in any case, even in the absence of it – which is limiting for this approach in the longer term.

The authors also apply the methodology in the formation of supramolecular gels. Much less data for this is presented in the paper, even though in the absence of the signal, this time a gel was not formed – so the difference between signal and no signal would appear (visually) to be larger than for polymer gels. Additional data to support this observation by rheology would be highly desirable and a larger degree of temporal control over materials performance in this system may strengthen the authors claims of temporal control (see previous paragraph).

I note that the title of the paper does not mention gels – indeed it is very general indeed – and as such, the paper may miss some of its target audience. I would suggest the authors include mention of the fact that the principles they have developed are exemplified in the triggered formation of gels.

I believe that the paper could be published in Nature Comm if the above aspects can be addressed. There is something conceptually interesting here, even if the system does not work quite as nicely/cleanly as might have been hoped. As such, I think other researchers will be inspired by the research and will move on to improve on the concept.

Reviewer #2 (Remarks to the Author):

In this work, Fanny Trausel and co-authors designed a deactivated aniline organocatalyst that is activated by the chemical signal hydrogen peroxide and catalyses hydrazone formation. The article is relatively valuable. However, the catalytic mechanism of pro-aniline 1 is not distinct. Values of innovation are not recommended enough. After reviewing this manuscript, we don't think this manuscript is suitable for nature communication.

We would like to thank the referees for their thorough evaluation of the manuscript, which will surely have a positive impact on its quality and readability.

Reviewers' comments:

Reviewer #1 (Remarks to the Author):

Remark:

The biggest issue with the manuscript is that, taken on its own the product of the H₂O₂-mediated reaction (compound 2) appears to be far less active as a catalyst (10 times) than the system formed by mixing its precursor (1) with H₂O₂. As such, it cannot really be concluded that compound 2 alone is responsible for the increase in reaction efficiency. The authors are up-front about this and state they cannot think of reasons why this may be the case.

Answer:

In the process of answering the referee, we discovered that the seemingly enhanced catalytic activity (vs aniline itself) of pro-aniline **1** with H₂O₂ is due to a side reaction or complexation between the NBD-hydrazide probe and pro-aniline **1**, leading to an increase of UV/vis absorbance at the same wavelength as the hydrazone product. This accounts for the perceived increased catalytic activity, but also (and more importantly) means that the NBD-hydrazone system is not suitable to show the activation of the pro-catalyst **1**. We thus set out to find a new hydrazone formation reaction that could show catalyst activation. The new hydrazone reaction we found (between 4-hydroxybutyric acid hydrazide **3** and 4-nitrobenzaldehyde **4**, Fig. 2a) is suitable to follow in UV-vis (Fig 2b), does not show any side reactions (Supplementary Table 1) and clearly shows the activation of pro-aniline **1** (Fig. 2c). As such, the remark of the referee has triggered a series of experiments that have surely enhanced the quality of the work and helped us avoid a serious error, and we would like to thank him/her for it.

We added the following to the manuscript:

We investigated the catalytic activity of pro-aniline **1** upon activation, by comparing the rates of hydrazone formation in a model reaction (Fig. 2a).^{25,26} Aldehyde **4** (0.5 mM) reacts with hydrazide **3** (0.1 mM) to form hydrazone **5** in a buffered medium (100 mM phosphate buffer pH 7.4) with 20% dimethylformamide (DMF) (Fig. 2b). This reaction is catalyzed by aniline **2** (0.5 mM), giving a 19-fold increase in reaction rate with respect to the uncatalyzed reaction. Unactivated pro-aniline **1** (0.5 mM) does not influence the reaction rate of hydrazone formation. Addition of H₂O₂ (2.5 mM) to pro-aniline **1** (0.5 mM) gives a relative reaction rate of 10, indicating efficient activation of the organocatalyst (Fig. 2b, Table 1). H₂O₂ alone (2.5 mM) does not increase the reaction rate of hydrazone formation (Supplementary Fig. 3a,b).

We replace Figure 2 and its caption by the following:

Figure 2: Control over the rate of hydrazone formation by activation of the catalyst. (a) Model hydrazone formation reaction, yielding the UV active probe **5**. Reaction conditions: 0.1 mM hydrazide **3**, 0.5 mM aldehyde **4**, 0.5 mM pro-aniline **1**, 0.5 mM aniline **2**, 2.5 mM (5 equivalents) H_2O_2 in 20% DMF in phosphate buffer (100 mM, pH 7.4). All experiments were carried out at 25 °C. The 20% DMF was used to ensure that **1** was completely dissolved. The rate of formation of **5** is 10 times higher when using H_2O_2 -activated pro-aniline **1** than for unactivated pro-aniline **1**. (b) Formation of **5** over time, without catalyst (black line), in the presence of aniline **2** (green line), in the presence of unactivated pro-aniline **1** (blue line) and in the presence of activated pro-aniline **1** with 5 equivalents of H_2O_2 (magenta line). (c) The rate of formation of **5** can be controlled during the process by adding a chemical signal (5 equivalents of H_2O_2 here after 1 hour), liberating the catalyst. Reaction with pro-aniline **1** and subsequent addition of H_2O_2 (magenta line), reaction with unactivated pro-aniline **1** (blue dashed line). After addition of the chemical signal an immediate response was observed: the reaction rate increased 9-fold. Table 1 Activation of the catalyst determines the initial rate of hydrazone formation. Pseudo-first-order reaction rates for hydrazone formation were determined by following the absorbance of **5** in UV/vis spectroscopy. The errors are the standard error of mean (the standard deviation divided by the square root of the number of measurements).

Remark:

I would like to see the authors outline more clearly why the aniline can act as a catalyst in the first instance – this is never clearly stated in the paper. This would help further considerations.

Answer:

Understanding the mechanism of catalysis by aniline **2** is indeed important for the paper. We added to the main text that ‘aniline is a nucleophilic catalyst for hydrazone formation’ and we refer to papers by Dirksen *et al.* and Bhat *et al.* that explain the use of aniline as a catalyst and its mechanism in great detail. We confirmed that aniline **2** is not used up during the hydrazone formation with a substoichiometric amount of pro-aniline **1** and H_2O_2 . After 65 h of reaction time, we were able to detect aniline **2** in the reaction mixture, using GC/MS. This result was added to the paper:

Although we confirmed complete conversion of pro-aniline **1** after addition of more than 1 equivalent of H_2O_2 and we were able to detect aniline **2** after an overnight hydrazone reaction in the presence of a substoichiometric amount of pro-aniline **1** + H_2O_2 (Supplementary Fig. 4), it might be the case that a small amount of aniline **2** is degraded over time.

Remark:

*The authors should then think carefully about the oxidising potential of hydrogen peroxide – does it oxidise the amine **2**, or form a complex with it – changing the nature of the catalyst in some way? Or*

does H₂O₂ play some other role in mediating the reaction between aldehyde and hydrazine (after all it does appear, on its own, to have a catalytic effect on the reaction).

Answer:

In order to ensure that H₂O₂ does not have an effect on the catalyst aniline **2**, we checked the absorbance of aniline before and after a night of exposure to H₂O₂ and the absorbance did not change for more than 5%. We also followed the hydrazone formation reaction in the presence of H₂O₂ and we found that it did not influence the reaction; we found the same behaviour as for the uncatalyzed reaction. In order to make this clearer in the paper, we added the following: 'H₂O₂ alone (2.5 mM) does not increase the reaction rate of hydrazone formation (Supplementary Fig. 3b).' Following the suggestion of the referee, we investigated the hydrazone reaction in the presence of both catalyst aniline **2** and H₂O₂, and we found the same behaviour as when we only use catalyst aniline **2** (Supplementary Table 1). We therefore conclude that, using our conditions and concentrations, H₂O₂ alone does not influence the reaction.

Supplementary Figure 3b: Absorbance at 350 nm of hydrazone **5** formation followed in UV/vis spectroscopy (conditions: hydrazide **3** (0.1 mM), aldehyde **4** (0.5 mM), 20% DMF in 100 mM phosphate buffer pH 7.4) without catalyst (blue line), with H₂O₂ (2.5 mM, red line). H₂O₂ has no influence on the reaction rate.

Remark:

Alternatively, does hydrogen peroxide change the pH of the medium? Were the reactions buffered, or pH controlled in some other way. All of these factors need considering before the authors could publish. I accept the problem may not be solvable, but it is important that the authors rule out as many possibilities as they can, and have some sort of proposal as to why they are observing this strange effect.

Answer:

It is indeed important that the pH remains constant during the reactions, especially because acid acts as a catalyst for hydrazone formation. We used buffered media for all reactions and we monitored the pH during the hydrazone reaction in the presence of pro-aniline **1** and H₂O₂. In order to make this clearer we added the following sentence to the main text: 'Furthermore, the pH was monitored during the reaction with pro-aniline **1** and H₂O₂: the pH remained stable at a value of 8. As our solvent system alone (100 mM phosphate buffer with 20% DMF) gives a pH of 8, we concluded that the reactions were sufficiently buffered.'

Remark:

The authors refer consistently to compound **2** as a 'catalyst'. Yet it is used in almost all examples at stoichiometric (or super-stoichiometric) levels. Often one of the other reagents is the limiting factor. Is it correct to strictly refer to it as a catalyst?? A catalyst must turn over multiple reactions, and I am

not sure I see evidence that it does. Again, if the authors were clear about the mechanism of aniline-mediated catalysis of these reactions, this may become clearer.

Answer:

We would like to refer to papers by Dirksen *et al.*, Bhat *et al.* and Larsen *et al.* in which the mechanism of aniline-catalysis is explained. Because aniline **2** is a relatively inefficient catalyst, it only affects the rate of reactions in an acceptable fashion when used in stoichiometric or super-stoichiometric amounts. This is also how it is used in many seminal studies (cf. Dirksen *et al.*; Bhat *et al.*). In order to show that aniline **2** is not used up during the reaction, we followed the hydrazone formation reaction with a substoichiometric amount of pro-aniline **1** + H₂O₂ overnight. The next day we were able to detect aniline **2** using GC/MS even though more than 1 equivalent of product had formed with respect to catalyst content (Supplementary Figure 4). We added to the paper: ‘we were able to detect aniline **2** after an overnight hydrazone reaction in the presence of a substoichiometric amount of pro-aniline **1** + H₂O₂ (Supplementary Fig. 4)’.

Supplementary Figure 4: GC/MS spectrum taken after 65 h of reaction time of hydrazone **3** with aldehyde **4** in the presence of pro-aniline **1** and H₂O₂. Conditions: hydrazone **3** (20 mM), aldehyde **4** (20 mM), pro-aniline **1** (10 mM), H₂O₂ (11 mM) in 20% DMF in 100 mM phosphate buffer pH 7.4. The reaction mixture was extracted after 65 h of reaction time with dichloromethane. The organic layer was evaporated and re-dissolved in ethyl acetate for GC/MS analysis. Aniline **2** was detected in the extract of the reaction mixture, MS (GC/MS) m/z: 93 [M], 66 [C₅H₆]⁺ (expected m/z = 93.06), retention time: 11 min.

Remark:

The authors apply the system in materials synthesis. I note that the difference made by the chemical trigger in the formation of the polymer gel is relatively small – after all, in the absence of hydrogen peroxide the gel still forms. As such, this system can only be seen as a first step towards what the authors are aiming for. I think they should perhaps describe what they observe as ‘partial temporal control’ – it is clear that the gelation is potentiated by the addition of the signal, but it would have happened in any case, even in the absence of it – which is limiting for this approach in the longer term.

Answer:

The referee read correctly that in the absence of the chemical signal and thus activation of the pro-catalyst, the polymer gel still forms. We would like to argue however, that we think the term ‘temporal control’ still applies, as the cross-over for G’ and G’’ (the gel point) for the catalysed gels is observed after 30 min, whereas this cross-over takes place after almost 2 hours for uncatalysed gels (Supplementary Fig. 5a).

Remark:

The authors also apply the methodology in the formation of supramolecular gels. Much less data for this is presented in the paper, even though in the absence of the signal, this time a gel was not formed – so the difference between signal and no signal would appear (visually) to be larger than for polymer gels. Additional data to support this observation by rheology would be highly desirable and a larger degree of temporal control over materials performance in this system may strengthen the authors claims of temporal control (see previous paragraph).

Answer:

We have put significant effort into providing rheological data of the formation of the supramolecular gel. However, because we had to add 20% methanol to this gel system in order to dissolve the pro-aniline, we were unable to obtain reproducible rheological results and hence, did not want to incorporate those in the manuscript. In response to the comment that the difference between signal and no signal appears larger for the supramolecular gels, we would like to point out that although the visual difference is larger, in both cases fibers were formed, which is in accordance with results described in Boekhoven *et al.*

Remark:

I note that the title of the paper does not mention gels – indeed it is very general indeed – and as such, the paper may miss some of its target audience. I would suggest the authors include mention of the fact that the principles they have developed are exemplified in the triggered formation of gels.

Answer:

We agree with the suggestion about the title and we decided to change to title to: ‘**Chemical signal activation of an organocatalyst enables control over soft material formation**’

Reviewer #2 (Remarks to the Author):

Remark:

[T]he catalytic mechanism of pro-aniline 1 is not distinct.

Answer:

Understanding the mechanism of catalysis by aniline is indeed important for the paper, but has already been discussed in significant detail by others. We added to the main text that ‘**aniline is a nucleophilic catalyst for hydrazone formation**’ and we refer to seminal papers by Dirksen *et al.* and Bhat *et al.* that explain the mechanism of aniline catalysis in great detail.

REVIEWERS' COMMENTS:

Reviewer #1 (Remarks to the Author):

The authors have carefully worked through my previous comments and addressed each of them carefully.

I still doubt whether accelerating gelation is truly 'temporal control' (which would indicate that a precise time for gelation could be dialled into the system).

Overall, however, I am very happy with the work the authors have done - in particular the way in which they have carefully and honestly uncovered some issues with their previous results and carried out new careful controls. This greater understanding has significantly improved the paper. I would be happy to recommend publication in Nature Comms.